# Differential Diagnosis of ICD-11 Personality Disorder and Autism Spectrum Disorder in Adolescents

**DOI:** 10.3390/children10060992

**Published:** 2023-06-01

**Authors:** Bo Bach, Martin Vestergaard

**Affiliations:** 1Psychiatric Research Unit, Center for Personality Disorder Research, Mental Health Services, Region Zealand, 4200 Slagelse, Denmark; 2Department of Psychology, University of Southern Denmark, 5230 Odense, Denmark; 3Department of Child and Adolescence Psychiatry (Copenhagen University Hospital), Mental Health Services, Region Zealand, 4000 Roskilde, Denmark

**Keywords:** ICD-11, autism, personality disorder, diagnosis, masking, female, women, classification

## Abstract

The International Classification of Diseases 11th Revision (ICD-11) introduces fundamentally new diagnostic descriptions for personality disorder and autism spectrum disorder. Instead of the traditional categorical taxonomies, both personality disorder and autism spectrum disorder are described as being on a continuum. Accumulating research has pointed out that, in some cases, adolescents with autism spectrum disorder are at risk of being confused with having a personality disorder, which particularly applies to female adolescents. Case reports describe how adult autistic women struggled with social and identity roles as children and adolescents, using compensatory strategies such as social imitation and other types of camouflaging. Furthermore, some adolescents with autism display emotion dysregulation and self-injury. The ICD-11 recognizes that features of autism spectrum disorder may resemble features of personality disorder, but the two diagnoses have not yet been formally compared to one another. The present article therefore sought to outline and discuss the overlap and boundaries between the ICD-11 definitions of personality disorder and autism spectrum disorder and propose guiding principles that may assist practitioners in differential diagnosis with female adolescents. We specifically highlight how aspects of the self and interpersonal functioning along with emotional, cognitive, and behavioral manifestations may overlap across the two diagnoses. Restricted, repetitive, and inflexible patterns of behavior, interests, and activities are core features of autism spectrum disorder, which may be masked or less pronounced in female adolescents. Collecting a developmental history of the early presence or absence of autistic features is vital for a conclusive diagnosis, including features that are typically camouflaged in females. A number of future directions for research and clinical practice are proposed.

## 1. Introduction

The co-occurrence, confusion, and misdiagnosis of personality disorder (PD) and autism spectrum disorder (ASD) have been increasingly investigated and discussed in recent years [1,2,3,4,5,6,7,8,9]. Both diagnoses are considered pervasive and persistent conditions, which affect all or most areas of psychosocial functioning. A recent meta-analysis of 22 studies on the co-occurrence of traditional PD categories and high-functioning ASD identified a mixed pattern of overlapping symptoms in PD and ASD that warrants more methodologically sound research [2].

The established ICD-10 diagnoses of autism, including Asperger’s syndrome, were not listed as official diagnoses until the release of DSM-III (1980) and ICD-10 (1994) based on the groundbreaking works by Leo Kanner and Lorna Wing [10]. Before this era, adolescents with autism would either be diagnosed with childhood schizophrenia or a residual diagnosis for a “borderline condition” if not simply neglected. Until a growing body of research pointed at biogenetic causes, it had been hypothesized that such autistic features were rooted in cold unemotional mothers referred to as “refrigerator mothers” [10,11]. In comparison, PD and borderline conditions were included already in ICD-6 (1949) and DSM-I (1952), influenced by Emil Kraepelin’s and Kurt Schneider’s definition of “psychopathic personalities” [12,13,14]. Thus, PD can be said to have an advantage over ASD, historically speaking.

With the ICD-11 classification, the traditional constructs of PD (e.g., Dependent, Avoidant, and Schizoid) are abandoned, as are the sub-classifications of autism (e.g., Childhood Autism, Atypical Autism, and Asperger’s syndrome). This calls for a reconsideration of the diagnostic boundaries and overlap between PD and ASD [15,16]. The ICD-11 explicitly recognizes that features of ASD may resemble features of PD and vice versa. As a general rule, when facing such issues with differential diagnosis, the ICD-11 points out that individuals with ASD should not be given an additional diagnosis of PD unless there are features of PD that cannot exclusively be accounted for by ASD and its co-occurring psychopathology such as eating disorder, OCD, non-suicidal self-injury, depression, or anxiety [17]. The question about the co-occurrence of PD and ASD also seems to be an emerging clinical challenge complicated by the introduction of a multi-axial system in DSM-III, in which the two diagnoses were allowed to co-occur [2,8,18]. In any case, as a neurodevelopmental disorder, the ICD-11 diagnosis of ASD generally overrules PD in the process of differential diagnosis.

Misdiagnosing ASD as PD may have serious consequences for the prognosis and self-concept of the individual [19,20]. Notably, misdiagnosing PD in individuals with ASD does not seem to be isolated to a few cases, as it was recently shown that a prior diagnosis of PD was among the most frequent mental disorders in adults diagnosed with ASD later in life [21]. A growing body of research in adults has shown that the clinical presentation of ASD in women is frequently confused with PD and other complex or co-occurring disorders [19,22,23,24,25,26,27,28,29]. These issues with differential diagnosis are likely to occur in adolescence, when the incidence for diagnosing ASD in girls increases steadily [30]. Thus, it seems important for clinicians not to get tunnel vision on symptoms of PD that may conceal underlying ASD in order to avoid yet another “lost generation” while also considering the risk of expanding the autism spectrum, resulting in potential overdiagnosis [31,32].

In the current article, we specifically focus on features and manifestations of PD and ASD in adolescents following the new diagnostic definitions in the ICD-11. When referring to adolescents, we use the WHO’s official definition in which an adolescent is someone between the ages of 10 and 19 years [33]. We recognize the significant variability of clinical manifestations along the ASD spectrum and highlight that the present paper particularly applies to what ICD-10 refers to as Asperger’s syndrome and Other Pervasive Developmental Disorder with the absence of intellectual impairment.

### Aim of the Present Article

Because the new PD diagnosis has not yet been formally compared to the new ASD diagnosis, the present article aimed to outline and discuss similarities and boundaries between definitions of the two ICD-11 diagnoses to provide guiding principles that may assist clinicians in diagnostic assessment. We specifically focus on implications for adolescents due to the importance of early detection and the anticipated underdiagnosis of girls with autism [34,35]. The comparison of diagnostic definitions is also expected to inform the development of hypotheses that may guide future empirical investigations. The following overview is faithfully based on definitions in the ICD-11 Clinical Descriptions and Diagnostic Requirements for both disorders [17]. Because research on the differential diagnosis of PD and ASD in adolescents is extremely limited, studies on adults is also taken into account. Based on our examination of the two sets of diagnostic descriptions, we have chosen to organize the following comparative overview according to aspects of the self, interpersonal functioning, emotional manifestations, cognitive manifestations, behavioral manifestations, developmental features, specific issues for female adolescents, and clinical assessment. Initially, the two new ICD-11 diagnoses will be defined and accompanied by a brief comparative historical overview.

## 2. The ICD-11 Classification of Personality Disorder

The ICD-11 classification of personality disorder (PD) abolishes all familiar PD types in favor of a global diagnosis indicating the presence of a PD followed by a classification of its severity (i.e., mild, moderate, or severe). The presence and severity of PD is determined by impairments in aspects of the self; interpersonal functioning; emotional, cognitive, and behavioral manifestations; and psychosocial impairment and distress [16]. Additionally, the clinician is allowed to specify up to five individual trait domains to capture the distinct “flavor” of the PD, which is beyond the scope of the present paper [17]. Table 1 presents an abbreviated version of the diagnostic features that determine the presence and severity of PD according to the ICD-11.

## 3. The ICD-11 Classification of Autism Spectrum Disorder

The ICD-11 classification of autism spectrum disorder (ASD), which is highly similar to the diagnosis of ASD in DSM-5, turns away from using the familiar categories of Childhood Autism, Atypical Autism, and Asperger’s syndrome in favor of a global spectrum of autism [10,15]. First of all, the ICD-11 requirements for an ASD diagnosis necessitate that the onset of the disorder occurs during the developmental period, typically in early childhood. However, the ICD-11 recognizes that the characteristic symptoms of ASD may not become fully manifested until later, when social demands exceed limited capacities (e.g., during early adolescence and sometimes adulthood). See Table 2 for an overview of developmental presentations.

The ASD diagnosis is defined by two components that must be present in terms of various potential expressions that may vary according to gender and chronological age: (1) persistent deficits in initiating and sustaining social communication and reciprocal social interactions that are outside the expected range of typical functioning given the individual’s age and level of intellectual development; (2) persistent restricted, repetitive, and inflexible patterns of behavior, interests, or activities that are clearly atypical or excessive for the individual’s age and sociocultural context. Notably, the second component also includes hypersensitivity and hyposensitivity to sensory stimuli, which were not considered diagnostic features in ICD-10 [15,17].

As with other mental and neurodevelopmental disorders, ASD symptoms must result in significant impairment in personal, family, social, educational, occupational, or other important areas of functioning. However, the ICD-11 recognizes that some individuals with ASD are able to function adequately in many contexts via exceptional effort, such that their deficits may not be apparent to others. The ICD-11 underscores that an ASD diagnosis is still appropriate in such cases because the clinical presentation may eventually occur in terms of accompanying decompensation symptoms such as depression, anxiety, and emotion dysregulation when social demands overwhelm the capacity to compensate [17]. According to ICD-11, this propensity particularly applies to adolescent girls [17], which is consistent with research indicating a diagnostic bias based on gender-specific adaptation skills that allow female patients to hide their social difficulties [36].

## 4. Aspects of the Self

According to the ICD-11, individuals with PD may suffer from a sense of identity that is either overly variable or excessively rigid. For example, they may not have a coherent sense of who they truly are, and therefore act and feel like a chameleon across various social settings [37,38].

Along the same lines, the ICD-11 recognizes that females with ASD are often able to function adequately in many contexts via exceptional effort in order to cover their underlying autistic vulnerabilities and difficulties. This is often done using compensating strategies such as masking, imitation, and camouflaging, which may also resemble and feel like the aforementioned “chameleon” behavior seen in PD [27,29,39]. In relation to gender-related identity, the ICD-11 also recognizes research showing that ASD is over-represented among children and adolescents with gender incongruence [40], which also applies to individuals with PD [41]. However, it is not entirely clear why a greater number of individuals with ASD and PD identify as transgender or gender-diverse and how gender incongruence is coupled to the sense of identity.

Whether ASD-related camouflaging in practice is confused with identity disturbance as seen in PD remains uninvestigated. However, some evidence suggests that the unstable self-image and disturbed sense of identity seen in PD might be something else than the identity problems seen in ASD [42]. Identity problems are highly prevalent in PD, usually experienced by the individual as having an unstable self-image and a disturbed identity [38,43,44]. In contrast, research on identity development in ASD is surprisingly sparse. Personal accounts from autistic women suggest that the experience of identity problems in adolescents may be related to their autistic features and mentalization problems [19]. Such personal descriptions include perceptions of having a “fragmented” identity or having an “absent” sense of self and mimicking other people’s behavior and style [45]. Studies specifically indicate that adolescents with autism may present with an “absent” or reduced psychological self due to impairments in self-referential encoding, autobiographical memory, and future thinking [46]. Autobiographical memories are central for our sense of self and have a directive function in guiding our values and beliefs, decision making, and goal-oriented behavior [47]. For instance, mentalizing and autobiographical memories are central to the development of narrative identity [44], and autistic adolescents experience problems with mentalizing and recalling autobiographical memories [48]. Thus, a lack of self-knowledge as seen in ASD [46] may cause a lack of self-directedness and self-sufficiency [49], which are core features of PD.

Moreover, like the low self-worth that often characterizes individuals with PD, low self-worth may also be expressed in many adolescents with ASD due to depressive symptoms and social defeat. Both social anxiety and mood disorders are common in ASD [50], and mood disorders are among the most frequent diagnoses prior to being diagnosed with ASD in adults [21].

Finally, the impaired capacity for self-direction in PD, such as unreasonable persistence in the pursuit of certain goals (e.g., anankastic personality), may in particular resemble ASD features such as a lack of adaptability to new experiences and circumstances, inflexible adherence to particular routines (e.g., only following familiar routes), persistent preoccupation with only one or more special interests, and excessive adherence to rules (e.g., when playing games).

Taken together, PD disturbances in identity, self-worth, and self-direction may be confused with similar manifestations in ASD. See the overview of PD and ASD in relation to aspects of the self in Table 3.

## 5. Interpersonal Functioning

Both PD and ASD are characterized by impaired interpersonal functioning, which may manifest in various ways. See the overview of PD and ASD in relation to interpersonal functioning in Table 4.

First, individuals with PD may have a compromised interest in engaging in relationships due to issues such as general avoidance, suspiciousness, or fear of rejection [17]. Similarly, individuals with ASD may show difficulty initiating and sustaining reciprocal social conversations, and their sensory overload may frequently cause them to withdraw [17]. In both cases, the dysfunctional patterns may lead to social withdrawal and isolation [18,51,52].

Second, individuals with PD may struggle with their ability to understand and appreciate others’ perspectives [17]. Such PD features are associated with hypermentalizing (reading too much into others’ behavior) as well as hypomentalizing (having problems with inferring the mental states of others). In a similar way, the ICD-11 points out that individuals with ASD may habitually find it difficult to imagine and respond to the feelings, emotional states, and attitudes of others, including verbal and non-verbal social communication [17]. For example, they may have an overtly literal understanding of others’ speech leading to inappropriate responses and conflicts that may resemble the conflict management problems seen in individuals with PD.

In comparison to mentalizing difficulties in PD, the diagnosis of ASD is mainly characterized by hypomentalizing, which is assumed to underlie the autistic social and communication impairments in autistic individuals [53]. However, if explicitly asked to mentalize, some individuals with ASD tend to hypermentalize like individuals with PD [54]. Thus, hypermentalizing and over-empathizing may be aspects of some autistic individual’s social compensation or hypersensitivity to social stimuli [27]. Overall, reading too much into others’ behavior is not unique to PD and does not exclude the presence of ASD.

Third, individuals with PD may have a compromised ability to develop and maintain close and mutually satisfying relationships (e.g., give-and-take relationships). In a similar manner, adolescents with ASD may exhibit a lack of mutual sharing of interest and have difficulty making and sustaining typical peer relationships [17]. While some individuals with PD may be submissive and self-victimizing in order to achieve others’ acceptance or avoid their rejection, the ICD-11 also recognizes that individuals with ASD may exhibit social naiveté, especially during adolescence, which can lead to exploitation by others as often reported by women with ASD [45].

Fourth, PD may sometimes be characterized by frantic attempts to maintain relationships in terms of not being abandoned by their significant other (e.g., a boyfriend). Such a pattern of what looks like separation anxiety often characterizes individuals with ASD [19,55] due to, for example, a lifelong and natural dependency on a parental figure who is the only person capable of interpreting the person’s autistic communication and needs [56,57,58]. This is often the case for mothers who naturally adapt to their child’s needs throughout upbringing while compensating for other people’s lack of understanding. Moreover, a frantic reaction to no longer having a familiar person in their life may also be attributed to the ASD-related distress caused by changes in routine and a lack of predictability. In other words, the frantic need for attachment seen in PD may sometimes be confused with a frantic need for sameness in ASD. Notably, some late-diagnosed females with ASD report a history of being passively “trapped” in intimate, and sometimes abusive, relationships due to a need of feeling accepted or following societal norms [45].

Finally, interpersonal dysfunction in PD may also be characterized by conflictual relationships, which is typically not associated with ASD. However, some individuals with ASD report a life-long history of having conflicts with others due to frequent misunderstandings [51]. Individuals with ASD may as children appear rude or antagonistic because others misread their behavior as intentionally provocative or externalizing [45]. Moreover, disruptive, impulsive, and conduct disorders have a high prevalence in ASD [50] and may be confused with PD-related antisocial behaviors. The same also applies to impaired social reciprocity and interpersonal conflicts driven by an autistic insistence on sameness, routine, and certain interests.

## 6. Emotional Manifestations

Impaired aspects of the self and interpersonal functioning in individuals with PD are typically manifested as emotional problems that may resemble ASD-related emotion regulation deficits [17]. See overview of PD and ASD in relation to emotional manifestations in Table 5.

First, PD may be characterized by emotional over-reactivity (e.g., affective hyperarousal on minor stressors), which may in some cases be attributed to hypersensitivity to sensory stimuli in ASD causing emotional turmoil [59]. The ASD-related emotion dysregulation is typically displayed as meltdowns or anger tantrums [60]. Case studies suggest that emotion dysregulation may comprise a major confounding issue when it comes to misdiagnosing PD in favor of ASD likely because the latter is not traditionally associated with emotion dysregulation [19]. However, accumulating evidence shows that ASD is associated with emotional dysregulation [61,62].

Second, PD may be characterized by impairments in range and appropriateness of emotional expression as well as emotional under-reactivity (e.g., flat or restricted affect), which sometimes may be attributed to ASD-related challenges with gestures, eye contact, facial expression, and body language [59]. Furthermore, low emotional awareness, recognition, and expression are seen in both PD [63] and ASD [59,64] and may therefore complicate differential diagnosis.

Third, individuals with ASD may, in general, experience emotional distress and aggressive outbursts in relation to specific triggers such as changes in routine, aversive sensory stimulation, unanticipated events, anxiety, and the interruption of rigid thought or behavior sequences [61]. For example, the experienced change in routine and “sameness” caused by the end of a relationship may invoke a “meltdown” with severe dysphoria and acute suicidality, which should not be confused with the abandonment-related desperation occasionally seen in individuals with PD.

Finally, the atypical sensory processing in ASD (i.e., hyposensitivity to interoceptive input such as pain or temperature) may also, in certain ways, resemble the PD-related inability to recognize and acknowledge emotions that are difficult or unwanted by the individual (i.e., alexithymia) [59,65].

## 7. Cognitive Manifestations

With respect to cognitive manifestations, PD may be characterized by the compromised accuracy of situational and interpersonal appraisals under stress, which may be evident in terms of psychotic-like perceptions. Along the same lines, the ICD-11 recognizes that disorganized thinking may also occur in ASD [17]. Meta-analytic evidence, including longitudinal research, shows that ASD is associated with an increased risk of psychosis [66,67]. In any case, ASD-related meltdowns may, in several ways, be comparable to the high affective arousal that causes impaired reality testing in PD.

PD may also be characterized by an impaired ability to make appropriate decisions in situations of uncertainty, which likely resembles the intolerance of uncertainty that naturally arises in ASD when engaging in unfamiliar experiences [68]. However, individuals with ASD may not only show difficulty with decision making in situations of uncertainty but also resistance to it.

Finally, PD may be characterized by problems with stability and flexibility in belief systems. Such issues may resemble rigidity in behavior and thinking as well as excessive adherence to rules that typically characterize individuals with ASD. Thus, in this case, PDs with prominent features of anankastia may be difficult to distinguish from ASD. The trait domain of anankastia is characterized by rigid, systematic, day-to-day routines with excessive scheduling and planning. Individuals with such PD features are inflexible and lack spontaneity, stubbornly insisting on following set schedules and adhering to plans, which is quite similar to ASD [3,4]. See overview of PD and ASD in relation to cognitive manifestations in Table 6.

## 8. Behavioral Manifestations

When it comes to behavioral manifestations, PD may be characterized by problems with flexibility and the modulation of behavior based on the situation (e.g., too much or too little restriction in behavior). In a comparable manner, ASD may sometimes involve explosive outbursts in response to an unanticipated change in routine, aversive sensory stimulation, or behavioral rigidity when the individual’s behavior sequences are interrupted (see Table 7). This pattern is consistent with research showing that disruptive behavior disorders, impulse control disorders, and attention-deficit hyperactivity disorders are common in ASD [50] as well as in PD [69,70].

In particular moderate–severe PD may often involve inappropriate behavioral responses to intense emotions and stressful circumstances in terms of self-harm [17]. Similarly, ASD may also involve self-injurious behaviors in response to decompensation and “meltdowns”. However, the ASD-related examples of self-harm, listed in the ICD-11, have more distinct and severe characteristics such as hitting one’s face and head banging [17], which are generally most common in cases of co-occurring intellectual impairment [71]. The significance of self-harm is consistent with a large-scale Swedish study (*N* = 410,732) that found ASD to have a fivefold increased relative risk of self-harm [72]. Notably, this risk was most pronounced for ASD without intellectual disability and particularly high for self-cutting and more violent methods, even when psychiatric co-morbidity was controlled for. Moreover, recent meta-analytic research also shows an increased risk of suicidal behavior in individuals with ASD [50,73]. Notably, a case study has illustrated how suicidality may also present as a restricted interest in ASD, which would not apply to PD [74].

## 9. Developmental Features

While many descriptive diagnostic features may overlap between PD and ASD, the assessment of developmental features seems to be key in the differential diagnosis. This is a core issue that aligns with the fact that ASD is a neurodevelopmental disorder. The ICD-11 is generally more flexible with diagnostic requirements for both PD and ASD when it comes to onset, stability, and developmental features (see Table 8).

First, it states that PD is “not typically” diagnosed in pre-adolescent children, while ASD per definition must be present in pre-adolescent children but is masked in many cases.

Second, it says that PD tends to appear first in childhood, increase during adolescence, and continue until fully manifest in adulthood, although individuals may not come to clinical attention until later in life. In a comparable manner, it says that the onset of ASD occurs during the developmental period, typically in early childhood, but characteristic symptoms may not become fully manifest until later, when social demands exceed limited capacities.

Third, in contrast to the ICD-10 definition, the ICD-11 states that PD is only relatively stable after young adulthood. This may be due to the fact that situational PD manifestations (e.g., emotion dysregulation, self-harm, and psychotic breaks) are more instable over time in comparison to stable features of underlying personality functioning and traits [75,76]. In a similar manner, the ICD-11 recognizes that a clinical presentation of ASD (e.g., manifest rigidity, striving for sameness, and withdrawal from social situations due to sensory overload) may not occur until social demands overwhelm the capacity to compensate. Accordingly, the ICD-11 points out that a first diagnosis of ASD in adulthood may be precipitated by a breakdown in domestic or work relationships, which might include an emerging pattern of “meltdowns”.

Fourth, the ICD-11 classification of PD exemplifies that self-harm and moodiness are more common during adolescence than during adulthood, which indicates that adolescence in general is a vulnerable life period. Similarly, for the same period, the ICD-11 points out that especially adolescent girls with ASD tend to withdraw socially and react with emotional changes to their social adjustment difficulties when depressive symptoms are a presenting feature.

Finally, the ICD-11 emphasizes that PD tends to arise when individuals’ life experiences provide inadequate support for typical personality development given the person’s temperament (i.e., the aspect of personality that is considered to be innate, reflecting basic genetic and neurobiological processes). Clinicians may therefore consider the presence of PD in cases when an individual reports childhood adversity. However, many girls growing up with unidentified ASD may have felt misunderstood and even mistreated by well-meaning parents and school teachers their entire life because the parental behavior was unfit for their ”hardwired” neurobiological characteristics.

## 10. Female Adolescents and the Role of Camouflaging

As presented in Table 9, the ICD classification addresses distinct manifestations and issues that apply to girls with autism rather than the traditional diagnostic definitions that relied on observations of boys [10,15].

Misdiagnosing PD for ASD appears to be more prevalent in women than men but seems to occur in both genders [21]. Interestingly, a cohort study of Danish citizens (*N* = 1.3 million) found the incidence of neurodevelopmental disorders to peak in late adolescence in girls, suggesting the possible delayed detection of such diagnoses in females [30]. This delayed diagnosis in girls seems critical as it has been shown that ASD diagnosis in older individuals is related to the accumulation of co-occurring psychopathology and distress [35,77].

In recent years, it has been more fully recognized that girls and women are under-represented if not absent in most ASD studies, which causes certain aspects of the scientific information to be less relevant for girls due to a biased focus on autistic features that are predominantly present in males [24,25,26]. For example, a “leaky” recruitment-to-research pipeline for females has been revealed in autism research. D’Mello and colleagues [25] found that males are four times as likely as females to be diagnosed with autism, which they attribute to a consistent pattern of enrolling only few females or none at all. Notably, the authors point out that this “leak” may be due to the Autism Diagnostic Observation Schedule (ADOS) for inclusion/exclusion. In their study, it was found that using the ADOS as a confirmatory diagnostic measure resulted in the mistaken exclusion of 2.5 times more autistic females than autistic males [25]. Instead of the undiagnosed ASD, the females are typically given other diagnoses such as PD. As portrayed by Darling Rasmussen [19], some misdiagnosed women eventually realize that they are not “broken” (i.e., mental disorder) but “just don’t fit in this world” (i.e., neurodevelopmental disorder).

A major reason for the under-diagnosis of ASD in females may be their increased use of masking in comparison to males with ASD [28]. Accordingly, one study found that autistic and non-autistic girls exhibited similar levels of social reciprocity seemingly due to compensatory camouflaging in autistic girls, while only autistic boys showed lover levels of reciprocity [27]. The authors suggest that this increased camouflaging in autistic girls may contribute to the delay of their diagnosis. Additionally, the ongoing use of camouflaging has indeed been found to compromise mental health, and, therefore, clinical presentations may naturally involve other mental health issues [39].

## 11. Clinical Assessment for Autism Spectrum Disorder in Females

Among 304 Swedish psychiatric outpatients that were consecutively screened and clinically assessed for ASD, the prevalence of an ASD diagnosis was estimated to at least 18.9%, with another 5–10% of patients having subthreshold symptoms [78]. These findings clearly suggest that ASD is underdiagnosed when the individual is no longer a child. Moreover, as already highlighted, the use of a golden standard instrument such as Autism Diagnostic Observation Schedule (ADOS) as a confirmatory diagnostic measure tends to result in the mistaken exclusion of ASD in females [25]. It therefore seems relevant with different approaches to clinical assessment that also bypass possible camouflaging in order to yield the necessary diagnostic information. For example, assessors should make sure to cover ICD-11-related hypersensitivity or hyposensitivity to sensory stimuli and compensating strategies (e.g., masking). See Table 10 for a list of proposed measures of clinician-rated instruments and self-report measures that may assist the assessment of ASD in adolescents. It is important to underscore that self-reports must only be used as a standardized screening that may serve as a foundation for subsequent dialogue and clinical assessment. In addition to the listed ASD instruments, it may also be informative to administer a personality inventory (e.g., Five-Factor Model) in order to characterize individual strengths and vulnerabilities that apply to all humans including individuals with ASD [79].

## 12. Clinical Assessment for Personality Disorder

When the assessment of ASD has been completed without any positive findings, it is a natural step to move on with an assessment for PD. In certain cases when an ASD diagnosis has been confirmed, it may also make sense to assess for PD when problems seem beyond what may be attributed to ASD (e.g., grandiose, reckless, impulsive, or antisocial attitudes or behavior). Instruments for the assessment of ICD-11-defined PD are still being developed and evaluated, and only two are currently available [86,87]. However, more well-established instruments developed for the DSM-5 Alternative Model of Personality Disorders (AMPD), such as STiP 5.1 [88], may be used to yield most of the information needed for diagnosing PD according to ICD-11. See Table 11 for a list of proposed measures of clinician-rated instruments and self-report measures that may assist the assessment of PD.

## 13. Conclusions

PD and ASD are pervasive and persistent conditions that may sometimes be confused with one another, particularly in female adolescents and adults. Both diagnoses share features related to aspects of the self; interpersonal functioning; and cognitive, emotional, and behavioral manifestations. This issue of resemblance or overlap is explicitly recognized in the most recent ICD-11 classification. The present paper sought to outline similarities and differences across the two diagnoses in order to highlight assessment and differential diagnosis.

ASD-related features such as persistent, restricted, repetitive, and inflexible patterns of behavior, interests, or activities with onset in early childhood are not characteristic features of PD. Nevertheless, it should be recognized that females with ASD tend to demonstrate fewer restricted, repetitive interests and behaviors than males, and they often make an exceptional effort to compensate for their symptoms during childhood, adolescence, or adulthood. Thus, they may be difficult to identify in a clinical setting. This is further complicated by girls’ tendency to withdraw socially and react with emotional changes to their social adjustment difficulties (e.g., display increased emotional reactivity). This presentation in girls may therefore mislead clinicians to focus on maladaptive personality functioning or co-occurring mental disorders (e.g., depression, eating pathology, OCD, anxiety) instead of the underlying and often camouflaged neurodevelopmental condition.

Because girls are so adept at camouflaging, distress generally only becomes manifest during mid-childhood and adolescence, when co-occurring mental disorders are misidentified as primary causes. Early mood difficulties often transform into more serious distress with emotional lability and self-harm, which is easily misrecognized as PD, causing preventable harm. Perhaps PD is something that many clinicians feel they can recognize immediately, increasing the need to consciously reflect on differential diagnoses, especially when presented with females who self-injure.

Based on what we have outlined in the present paper, we specifically hope for the following five improvements within clinical care, training, and research: (1) the differential diagnoses of ASD and PD in adolescence within the training curriculum of mental health professionals; (2) a greater understanding of the co-occurrence of mental health issues, which complicates the differential diagnoses of ASD and PD; (3) the study of more representative clinical populations that include adequate sample sizes for females and males in studies on ASD as well as PD; (4) the further understanding of and differentiation between aspects of social communication across the two diagnoses in females; and (5) an increase in clinical expertise in detecting ASD early in females.

It is also our hope that future research, in particular within the field of PD research, will pay more attention to the issues outlined in this article.

Finally, we also welcome investigations of adapted treatment models for problems that often accompany ASD (e.g., emotion dysregulation, self-harm, poor self-worth, and interpersonal difficulties), which may apply to recent adaptions of dialectical behavior therapy [99,100,101], mentalization-based treatment [102,103], and schema therapy [104,105] for autistic people.

## Figures and Tables

**Table 1 children-10-00992-t001:** Abbreviated presentation of aspects that are used to determine the presence and severity of a personality disorder according to ICD-11.

**Personality Functioning**			
**Aspects of the self**		**Interpersonal functioning**	
Identity Self-worth Accuracy of self-view Self-directedness		Engagement in relationships Perspective-taking Mutuality Conflict management	
**Manifestations**			
**Emotional**	**Cognitive**		**Behavioral**
Emotional experience Emotion regulation Recognizing emotions	Reality testing Decision making Rigidity and flexibility		Impulse control Harm to self Harm to others
**Psychosocial impairment and/or distress**			

**Table 2 children-10-00992-t002:** Developmental presentations of autism spectrum disorder from middle childhood to adulthood according to ICD-11.

*Middle Childhood*: In children with ASD without a disorder of intellectual development, social adjustment difficulties outside the home may not be detected until school entry or adolescence when social communication problems lead to social isolation from peers. Resistance to engage in unfamiliar experiences and marked reactions to even minor changes in routines are typical. Furthermore, excessive focus on detail as well as the rigidity of behavior and thinking may be significant. Symptoms of anxiety may become evident at this stage of development. *Adolescence*: By adolescence, the capacity to cope with increasing social complexity in peer relationships at a time of increasingly demanding academic expectations is often overwhelmed. In some individuals with ASD, the underlying social communication deficits may be overshadowed by the symptoms of co-occurring mental and behavioral Disorders. Depressive symptoms are often a presenting feature. *Adulthood*: In adulthood, the capacity for those with ASD to cope with social relationships can become increasingly challenged, and clinical presentation may occur when social demands overwhelm the capacity to compensate. Presenting problems in adulthood may represent reactions to social isolation or the social consequences of inappropriate behavior. Compensation strategies may be sufficient to sustain dyadic relationships but are usually inadequate in social groups. Special interests, and focused attention, may benefit some individuals in education and employment. Work environments may have to be tailored to the capacities of the individual. A first diagnosis in adulthood may be precipitated by a breakdown in domestic or work relationships. In ASD, there is always a history of early childhood social communication and relationship difficulties, although this may only be apparent in retrospect.

Note. Adapted from ICD-11 Clinical Descriptions and Diagnostic Requirements for Autism Spectrum Disorder. A complete presentation of developmental manifestations, including infancy, is available in the original source [17].

**Table 3 children-10-00992-t003:** Comparative overview of personality disorder and autism spectrum disorder in relation to aspects of the self.

Personality Disorder	Autism Spectrum Disorder
Instability and incoherence of one’s sense of identity (e.g., extent to which identity or sense of self is overly variable and inconsistent or overly rigid and fixed).	Some individuals with ASD are able to function adequately in many contexts via exceptional effort, such that their deficits may not be apparent to others. They may use compensation strategies (e.g., masking), resulting in a ”chameleon” identity.
Inability to maintain an overall positive and stable sense of self-worth.	During adolescence and adulthood, depressive disorders are often a presenting feature, which include low self-worth.
Incapacity for self-direction (e.g., unreasonable persistence in pursuit of certain goals).	Lack of adaptability to new experiences and circumstances, inflexible adherence to particular routines (e.g., only following familiar routes), persistent preoccupation with one or more special interests, and excessive adherence to rules (e.g., when playing games).

**Table 4 children-10-00992-t004:** Comparative overview of personality disorder and autism spectrum disorder in relation to interpersonal functioning.

Personality Disorder	Autism Spectrum Disorder
Compromised interest in engaging in relationships with others.	Problems with understanding and use of language in social contexts and ability to initiate and sustain reciprocal social conversations.
Compromised ability to understand and appreciate others’ perspectives.	Difficulties with imagining and responding to the feelings, emotional states, and attitudes of others; misunderstands or shows inappropriate responses to the verbal or non-verbal social communications of others; inadequate social awareness leading to behavior that is not appropriately modulated according to the social context; has an overly literal understanding of others’ speech.
Inability to develop and maintain close and mutually satisfying relationships.	Problems with mutual sharing of interests; the ability to make and sustain typical peer relationships; social naiveté, especially during adolescence, leading to exploitation by others.
Inability to manage conflict in relationships.	Pedantic precision and insistence on sameness with a lack of compromising.

**Table 5 children-10-00992-t005:** Comparative overview of personality disorder and autism spectrum disorder in relation to emotional manifestations.

Personality Disorder	Autism Spectrum Disorder
Issues with range and appropriateness of emotional experience and expression.	Integration of spoken language with typical complimentary non-verbal cues, such as eye contact, gestures, facial expressions, and body language. These non-verbal behaviors may also be reduced in frequency or intensity.
Tendency to be emotionally over-reactive.	Lack of adaptability to new experiences and circumstances, with associated distress, that can be evoked by trivial changes to a familiar environment or in response to unanticipated events. Lifelong excessive and persistent hypersensitivity or hyposensitivity to sensory stimuli. Disruptive behavior with aggressive outbursts (e.g., explosive rages) often associated with a specific trigger (e.g., a change in routine, aversive sensory stimulation, anxiety, or rigidity when the individual’s thoughts or behavior sequences are interrupted).
Tendency to be emotionally under-reactive and compromised ability to recognize and acknowledge emotions that are difficult or unwanted by the individual (e.g., anger and sadness).	Excessive hyposensitivity to sensory stimuli along with features of alexithymia.

**Table 6 children-10-00992-t006:** Comparative overview of personality disorder and autism spectrum disorder in relation to cognitive manifestations.

Personality Disorder	Autism Spectrum Disorder
Compromised accuracy of situational and interpersonal appraisals, especially under stress (i.e., impaired reality testing).	Disorganized thinking and behavior or psychotic episodes may occur in response to “breakdowns” and “meltdowns”.
Inability to make appropriate decisions in situations of uncertainty.	Situations of uncertainty may be influenced by resistance to engage in unfamiliar experiences.
Reduced flexibility of belief systems.	Excessive adherence to rules and rigidity of thinking.

**Table 7 children-10-00992-t007:** Comparative overview of personality disorder and autism spectrum disorder in relation to behavioral manifestations.

Personality Disorder	Autism Spectrum Disorder
Compromised flexibility in controlling impulses and modulating behavior based on the situation and consideration of the consequences.	Disruptive behavior with aggressive outbursts (e.g., explosive rages) may be a prominent feature of ASD when it is often associated with a specific trigger (e.g., a change in routine, aversive sensory stimulation, anxiety, or rigidity when the individual’s thoughts or behavior sequences are interrupted).
Problematic behavioral responses to intense emotions and stressful circumstances (e.g., propensity to self-harm or violence).	Self-injurious behaviors (e.g., not only hitting one’s face and head banging but also “cutting” as seen in PD).

**Table 8 children-10-00992-t008:** Comparative overview of personality disorder and autism spectrum disorder in relation to developmental features.

Personality Disorder	Autism Spectrum Disorder
Features of PD are not typically diagnosed in pre-adolescent children.	Features of ASD must be present in pre-adolescent children.
Features of PD tend to appear first in childhood, increase during adolescence, and continue to be manifest into adulthood, although individuals may not come to clinical attention until later in life.	Onset of ASD occurs during the developmental period, typically in early childhood, but characteristic symptoms may not become fully manifest until later, when social demands exceed limited capacities.
PD is relatively stable after young adulthood.	A first ASD diagnosis in adulthood may be precipitated by a breakdown in domestic or work relationships. Clinical presentation may not occur until social demands overwhelm the capacity to compensate.
Self-harm and moodiness are more common during adolescence than during adulthood.	Girls with ASD tend to withdraw socially and react with emotional changes to their social adjustment difficulties. Depressive symptoms often comprise a presenting feature.

**Table 9 children-10-00992-t009:** Features in females with autism spectrum disorder according to ICD-11.

Males are four times more likely than females to be diagnosed with ASD. Those females who are diagnosed with ASD are more frequently diagnosed with co-occurring disorders of intellectual development, suggesting that less severe presentations may go undetected compared to males. Females tend to demonstrate fewer restricted, repetitive interests and behaviors than males. Some individuals with ASD are capable of functioning adequately by making an exceptional effort to compensate for their symptoms during childhood, adolescence, or adulthood. Such sustained effort, which may be more typical of affected females, can have a deleterious impact on mental health and well-being. During middle-childhood, boys may act out with reactive aggression or other behavioral symptoms when challenged or frustrated, while girls tend to withdraw socially and react with emotional changes to their social adjustment difficulties.

**Table 10 children-10-00992-t010:** Suggested approaches to assessment for autism spectrum disorder in adolescents.

*Clinician-Rated Instruments* − Autism Diagnostic Observation Schedule 2 (ADOS-2) [80], Module 4 − Developmental, Dimensional and Diagnostic Interview—Adult Version (3Di-Adult) [81,82] *Self-Report Measures* − Ritvo Autism Asperger Diagnostic Scale–Revised (RAADS-R) [83] − Camouflaging Autistic Traits Questionnaire (CAT-Q) [84] − Girls Questionnaire for Autism Spectrum Condition (GQ-ASC) [85]

**Table 11 children-10-00992-t011:** Suggested approaches to assessment for personality disorder in adolescents.

*Clinician-Rated Instruments* − Semi-Structured Interview for DSM-5 Personality Functioning (STiP 5.1) [88,89,90] − Structured Clinical Interview for DSM-5 Alternative Model of Personality Disorder (SCID-5-AMPD), Module I [91] − Personality Disorder Severity–ICD-11 (PDS-ICD-11) Clinician-Rating Form [92] *Self-Report Measures* − Clark et al.’s Preliminary Scales for Personality Functioning [87] − Personality Disorder Severity–ICD-11 (PDS-ICD-11) Patient-Report Form [86] − Level of Personality Functioning–Brief Form (LPFS-BF) [93,94,95] − Level of Personality Functioning Questionnaire–12-18 (LoPF-Q-12-18) [96,97] − Self- and Interpersonal Functioning Scale (SIFS) [98]

## Data Availability

No new data were created or analyzed in this study. Data sharing is not applicable to this article.

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
