# Peer review of "Differential Diagnosis of ICD-11 Personality Disorder and Autism Spectrum Disorder in Adolescents"

_children, 2023, doi:10.3390/children10060992_

Round 1

Reviewer 1 Report

The article discusses an interesting topic in clinical practice and research related to autism spectrum disorder and offers fresh information in comparison to earlier publications of a similar nature. However, it still requires significant improvements, particularly in the descriptions of the methods that were employed to achieve the goal of offering guiding principles that could help clinicians with the diagnostic procedure. Furthermore, the report appears to refer to a very specific minority fraction of individuals who fall within the broad ASD spectrum without offering any justification.

A detail of suggested changes is provided with notes to the attached PDF.

Author Response

REVIEWER 1

The article discusses an interesting topic in clinical practice and research related to autism spectrum disorder and offers fresh information in comparison to earlier publications of a similar nature. However, it still requires significant improvements, particularly in the descriptions of the methods that were employed to achieve the goal of offering guiding principles that could help clinicians with the diagnostic procedure. Furthermore, the report appears to refer to a very specific minority fraction of individuals who fall within the broad ASD spectrum without offering any justification.

RESPONSE:
We appreciate this feedback and all the comments provided in the pdf. Please find attached the pdf.-file in which we respond to each comment (pointing at revisions in the manuscript).

Reviewer 2 Report

Dear authors,

thanks for your work on differential diagnosis of Personality Disorders and ASD that is highly relevant in clinical practice and often challenging, particularly in women.

There are a few major suggestions and a number of minor corrections I would like to make to improve the clarity and contents of your manuscript:

-The comparative overviews you provide in tables 3-8 could be much improved and need revision:

-Avoid too much text in the tables as you explain the similarities and differences in the manuscript anyway).

-In addition, headings/descriptions of the "lines" of the tables would be most helpful for practical use of the tables in clinical work: eg. for table 4: something like: interest in social relationships, perspective taking (ToM) , developing and maintaining social relationships, conflict management.

-In the columns of your tables I recommend to describe the symptomataology e.g. table 4: lack of understanding, or limited ability/disability, problems in mutual sharing of interests etc. as well as strengths rather than using general categories only

-I would like to encourage you to elaborate (particularly in chapter 5, and 6, probably also ch. 7) the discussion of differences in Social Communication that is certainly the major definition criterion for ASD. This difference should also be considered in your conclusion and abstract. I do not think that your conclusions in the abstract (lines 26-28) with a focus on differences in RBB (rather than Social Communication) can be derived from your whole manuscript and the currents state of the literature. Social communication characteristics should be considered a key in differential diagnosis of PD and ASD.

-In your introduction I would like to see more empirical data eg quantitative data on symptoms of PD in ASD, for example from the empirical studies included in Rinaldi et al's review (you reference in your paper).

-I miss a dicussion on gender dysphoria that is a very frequent topic in clinical practice!!!

-Please provide more information on the self-reports  (table 10) suggested for ASD diagnosis in adolescents. 

-I would be interested in your clinical recommendations concerning the assessment and description of atypical personality traits in patients with ASD that (even without the diagnosis of PD as a comorbidity) can be highly relevant for understanding the patient and for treatment.

-The paper would certainly benefit from case vignettes (eg. ASD without personality disorder and ASD with Personality Disorder) .

typos:

line 37  affect instead of affects

line 164 females instead of Females

table 1: experience instead of experiecnce

There are a few instances of (REF) eg. line 446 in your MS that need to be specified.

A final English check is recommended.

Author Response

Please find responses attached.
